# ACTIVATION FUNCTION MATTERS IN GRAPH TRANS-FORMERS

## ABSTRACT

Following the success of Transformers in deep learning, Graph Transformers have emerged as one of the most prominent architectures for graph representation learning. At the heart of Graph Transformers lies the self-attention mechanism, which aims to elevate information from nodes similar to the query node while suppressing information from others. However, this paper has unveiled a critical limitation: the attention mechanism in MPNN-based Graph Transformers cannot effectively discern the number of neighbors, resulting in a restricted expressive capacity. To address this limitation, we investigate three activation functions for the attention mechanism: $softmax$, $tanh$, and $sigmoid$, and show that $sigmoid$ is the most powerful. Our study culminate in the development of an enhanced variant of the Graph Transformer, known as the Expressive Graph Transformer (EGT), and we prove that EGT effectively distinguish number of neighbors without restricted expressive capacity. Extensive evaluations on graph classification and node classification demonstrate the effectiveness and robustness of EGT. Our code is released at `https://anonymous.4open.science/r/EGT-98CA/`.

## 1 INTRODUCTION

Graphs are widely applicated for representing relations between entities, such as social networks (Leskovec et al., 2008) and protein networks (Borgwardt et al., 2005). In the realm of graphs, entities and their interconnections are abstracted as nodes and edges, respectively. As efficient approaches to model information in graph data, Graph neural networks (GNNs) (Gori et al., 2005; Duvenaud et al., 2015) have witnessed a surge in popularity over recent years as efficient methodologies for modeling information within graph data.

A prevalent technique for modeling graphs is the message-passing algorithm, wherein nodes iteratively accumulate information from their neighbors (Kipf & Welling, 2017) and the aggregation is usually permutation-invariant. The GNN which follows message passing paradigm are commonly referred to as message passing neural networks (MPNNs). Research efforts in the domain of MPNNs mainly focus on the way of each node aggregating information (Veličković et al., 2018; Hamilton et al., 2017; Brody et al., 2022) and sampling mechanism (Chen et al., 2018; Zeng et al., 2019) of its neighbors. In particular, Xu et al. (2019) utilize Weisfeiler-Lehman (WL) test (Weisfeiler & Leman, 1968) to explore the expressive power of GCN and GraphSAGE and propose GIN to achieve WL test. However, the expressiveness of attentive MPNNs has not been investigated.

Recently, as Transformer (Vaswani et al., 2017) achieves success in natural language processing since its superiority in fitting data in Euclidean space and capturing interaction, Graph Transformers (Yun et al., 2019; 2020; Hu et al., 2020b) are proposed to model information in graphs and become one of the most popular GNN architectures. Specifically, Dwivedi & Bresson (2021) propose GT adhering to the message passing scheme, which outperforms traditional MPNNs. Inspired by Transformer, most Graph Transformers ignore the selecting of attentive activation function and adopt $softmax$. Nevertheless, we observe that use inappropriate activation function (such as $softmax$) lacks the capacity to distinguish the number of neighbors, resulting in the loss of crucial information and limited expressive power.

In this work, we propose theoretical derivations to illustrate that MPNN-based Graph Transformers (Yun et al., 2019; Hu et al., 2020b; Dwivedi & Bresson, 2021) suffer from constrained expressive capacity and inadequacy in passing the WL test. Our key insight is that the non-injective attentive acti-

vation function lead to restricted expressivity. Specifically, the widely used *softmax* is non-injective, thereby limiting the expressive power of mainstream MPNN-based Graph Transformers. Additionally, we delve into studying two potentially competitive activation functions, $tanh$ and $sigmoid$, and provide evidence to substantiate the superiority of $sigmoid$ as the most potent activation function among them.

Based on these observations, we develop a variant of Graph Transformer with high expressive power, namely Expressive Graph Transformer (EGT), which is proved to be as powerful as WL test. EGT leverages the $sigmoid$ activation function in the attention mechanism and incorporates improvements in feed-forward layers. Our primary contributions can be summarized as follows:

- We show that the MPNN-based Graph Transformers cannot achieve the same level of expressive power as the WL test.
- We conduct an in-depth exploration of the expressivity of attentive functions and identify specific graph structures that cannot be effectively distinguished by both the *softmax* and *tanh* activation functions.
- We propose an improved architecture of Graph Transformer and show that its expressive power is equal to the power of the WL test.
- Our extensive experimental evaluations, spanning graph classification and node classification tasks, show the efficiency of our model and its ability to significantly outperform state-of-the-art methods. Additionally, analyses focusing on noise and hyperparameters demonstrates the effectiveness and robustness of EGT.

## 2 RELATED WORK

**Attention in GNNs**. Attention in GNNs modeling pairwise interactions between elements in graph-structured data can be traced back to interaction networks (Battaglia et al., 2016) and relational reasoning (Santoro et al., 2017). A notable work is GAT (Veličković et al., 2018), which learns from Transformer and is a simple and general framework with attention. However, Brody et al. (2022) recently discover that GAT is limited to static attention and introduced GATv2 to enable dynamic attention, addressing this constraint.

**Graph Transformer**. Various Graph Transformers have been proposed for node classification and graph classification tasks. For node classification, Graph Transformers usually adopt the message passing structure that enables them to handle large graphs effectively (Yun et al., 2019; Hu et al., 2020b; Dwivedi & Bresson, 2021). Recent efforts have predominantly focused on enhancing computational efficiency (Wu et al., 2022; Chen et al., 2023) and structural embedding (Mao et al., 2023). In the context of graph classification, Graph Transformers resemble traditional Transformers, employing attention mechanisms to capture interactions between pairs of nodes. These approaches utilize structural encodings, also referred to as biases, to incorporate connectivity information (Ying et al., 2021; Zhang et al., 2023).

**Expressive capacity of MPNNs.** Since the seminal work (Xu et al., 2019) discussing the expressive power of GCN and GraphSAGE, and proposing GIN which matches the expressiveness of the WL test, numerous works studies high-order expressivity (Morris et al., 2019; Maron et al., 2019; Bevilacqua et al., 2022) by subgraph MPNNs. Wijesinghe & Wang (2021) extended MPNNs beyond the WL test by calculating connectivity within subgraphs. However, the expressive capacity of attentive MPNNs, such as GAT and GT, remains unexplored.

## 3 PRELIMINARY

A graph $\mathcal{G}$ can be defined as a tuple $(\mathbb{V}, \mathbb{E})$, where $\mathbb{V}$ represents the set of nodes within $\mathcal{G}$, and $\mathbb{E}$ denotes the set of edges. The connectivity of the graph is represented by the edges where each entry $e_{u,v}$ indicates the presence of an edge connecting nodes $u$ and $v$. For a given node $u \in \mathbb{V}$, its neighbors can be denoted as $\mathcal{N}(u) = \{v \in \mathbb{V} \mid e_{u,v}\}$.

The Weisfeiler-Lehman test (Weisfeiler & Leman, 1968), referred to as WL test for notation simplicity, is a well-established algorithm utilized to determine the isomorphism between two graphs.

The WL algorithm maintains a state for each node that is iteratively refined by aggregating the states of its neighboring nodes. In details, it hashs the multiset of the states into unique states, which can be represented as

$$c^t(u) = \text{HASH}\Big(c^{t-1}(u), \{\!\{c^{t-1}(v) \mid v \in \mathcal{N}(u)\}\!\}\Big). \tag{1}$$

The WL algorithm continues to refine the states of the neighbors until $c^t(u) = c^{t-1}(u)$ for all nodes $u \in \mathbb{V}$. It can be applied to test the expressiveness of GNNs as well. The aggregating procedure is similar to MPNNs: for each node, it aggregate and calculate the neighbors iteratively, and the aggregated structure looks like a rooted subtree.

For the purpose of embedding graphs, the predominant approach involves the utilization of message passing mechanisms to aggregate information from neighboring nodes, such as GCN (Kipf & Welling, 2017), GAT (Veličković et al., 2018), etc. Here we reformulate MPNNs as follows:

$$
\begin{aligned}
\boldsymbol{a}^t(u) &= \text{AGGREGATE}^t\Big(\{\!\{\boldsymbol{f}^{t-1}(v) \mid v \in \mathcal{N}(u)\}\!\}\Big), \\
\boldsymbol{f}^t(u) &= \text{COMBINE}^t\Big(\boldsymbol{f}^{t-1}(u), \boldsymbol{a}^t(u)\Big),
\end{aligned}
\tag{2}
$$

where $\boldsymbol{f}^t(u)$ denotes the feature vector of node $u$ at $t$ layer and $\{\!\{\cdot\}\!\}$ is the multiset. AGGREGATE$^t$ and COMBINE$^t$ are custom functions in GNNs. For GT, a standard Graph Transformer, the AGGREGATE$^t$ and COMBINE$^t$ are

$$
\begin{aligned}
\text{AGGREGATE}^t &= \overset{H}{\|} \Big( \sum_{v \in \mathcal{N}(u)} \boldsymbol{w}_v^t(u)\boldsymbol{v}_v^t(u)\Big), \\
\text{COMBINE}^t &= \text{FFN}^t\big(\boldsymbol{f}^{t-1}(u) + \boldsymbol{a}^t(u)\big),
\end{aligned}
\tag{3}
$$

where FFN$^t(\cdot)$ denotes the feed-forward layers including non-linear activation function, normalization and dropout. $\|$ signifies matrix concatenation. The aggregation function employs multi-head attention, treating the node itself as *queries*, while considering its neighbors as *keys* and *values*. Specifically, $\boldsymbol{v}_v^t(u) = \boldsymbol{W}_V^t \boldsymbol{f}^{t-1}(v)$ is the linear transformation of $\boldsymbol{f}^{t-1}(v)$, which denotes the representation of neighbors from the previous aggregation iteration, with $W_v^t$ being a learnable matrix. $\boldsymbol{w}_v^t(u)$ is the attention weight of the node $v$, one neighbor at a $t$-hop distance from node $u$, that can be computed by

$$\boldsymbol{w}_v^t(u) = softmax\Big(\frac{\boldsymbol{q}^t(u)\boldsymbol{k}_v^{t\top}(u)}{\sqrt{d_h}}\Big). \tag{4}$$

where $\boldsymbol{q}^t(u) = \boldsymbol{W}_Q^t \boldsymbol{f}^{t-1}(u)$, $\boldsymbol{k}_v^t(u) = \boldsymbol{W}_K^t \boldsymbol{f}^{t-1}(v)$ are the *queries* and *keys* of the attention. $d_h$ is the number of attention heads.

## 4 THE EXPRESSIVE POWER OF GRAPH TRANSFORMER

Following the remarkable success of Transformers, attention mechanisms have been extensively employed in MPNN-based Graph Transformers. However, most of them use non-injective activation function in attention mechanism (*softmax*) leading to the constrained expressive capacity which is lower than WL test. In our study, we perform analyses involving two alternative attention variants: the utilization of the *tanh* and *sigmoid* activation functions in place of *softmax*. Surprisingly, we observe that both *softmax* and *tanh* encounter difficulties in distinguishing remarkably simple graphs, underscoring their inferiority. We provide theoretical proofs and engage in a comprehensive discussion of their implications. Additionally, we introduce an expressive graph transformer for more proficient modeling of nodes and their neighbors.

### 4.1 EXPRESSIVE CAPACITY OF SOFTMAX

The *softmax* function is the predominant choice as the activation function to compute attention coefficients. Through $softmax(x) = \dfrac{\exp(x_i)}{\sum_j \exp(x_j)}$, the *keys* similar to the *query* are amplified, while the others are suppressed. This guarantees the attention coefficients of all *keys* are positive

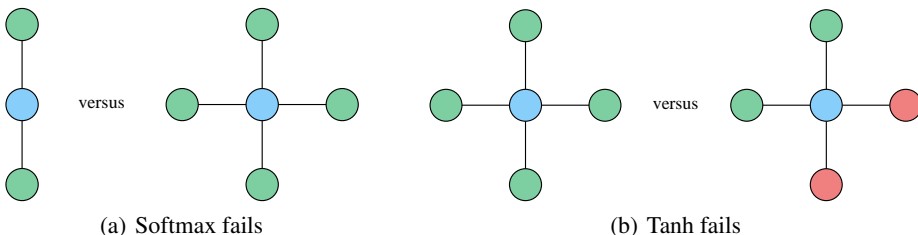

(a) Softmax fails                    (b) Tanh fails

Figure 1: Examples of graph structures that softmax and tanh activation functions fail to discriminate. Between the two graphs, the target nodes (depicted as blue nodes) get the same embedding despite the disparities in their graph structures.

and sum up to 1. However, due to this characteristic, it lacks the ability to discern the number of neighbors, as it is not sensitive to the count of *keys* when calculating coefficients.

In Figure 1(a), when presented with a target node and its neighbor nodes bearing identical labels, *softmax* assigns equal weights to all the neighbors. Consequently, when aggregating the weighted contributions of the neighbors, the outcomes remain identical, irrespective of the number of neighbors. This observation intuitively illustrates that *softmax* is not an injective attentive function.

**Theorem 1.** The expressive capacity of the MPNN-based Graph Transformers using *softmax* as an attentive function is inferior to that of the WL test.

Proofs of all Theorems and Corollaries can be found in the Appendix. Nonetheless, it is important to acknowledge that *softmax* retains its effectiveness when the number of computed components remains constant, as seen in applications like image or natural language processing. In such cases, it can efficiently compress component representations, regardless of their quantity, rendering it well-suited for multi-layer attention stacking in Transformers. Neighbor sampling mechanisms can be employed to mitigate the associated limitations.

### 4.2 EXPRESSIVE CAPACITY OF HYPERBOLIC TANGENT

What happens if we replace *softmax* with hyperbolic tangent function in the attention mechanism? $tanh(x) = \dfrac{e^x - e^{-x}}{e^x + e^{-x}}$ maps any value into the interval $(-1, 1)$ and it is also permutation-invariant. Most importantly, the mapping procedure is injective. Nevertheless, when applied to the AGGREGATE operation in Graph Transformer, *tanh* loses its injectivity.

In Figure 1(b), considering that the neighbors of the target node have opposite values, *tanh* assigns opposite weights to these neighbors as well. Specifically, taking the simplest matrix transformation where $\boldsymbol{k}_v^1(u) = \boldsymbol{v}_v^1(u) = \boldsymbol{f}^0(v)$ as an example, we notice that $\boldsymbol{w}_v^1(u)\boldsymbol{v}_v^1(u) = \boldsymbol{w}_{v'}^1(u)\boldsymbol{v}_{v'}^1(u)$ holds true for any $v$ and $v'$, regardless of the sign of their values.

**Theorem 2.** The expressive capacity of MPNN-based Graph Transformer using *tanh* as attentive function is inferior to that of the WL test.

In contrast to *softmax*, *tanh* exhibits sensitivity to the number of neighbors. One approach to address this limitation is by introducing an additional dimension to indicate whether the input is positive. However, this approach suffers from poor scalability and is unsuitable for multi-layer MPNNs.

### 4.3 EXPRESSIVE CAPACITY OF SIGMOID

Utilizing *sigmoid* as the activation function for attention has the potential to be injective when considering information from neighbors. It possesses the ability to distinguish both the number and sign of neighbors. Moreover, $sigmoid(x) = \dfrac{1}{1 + \exp(-x)}$ scales and confines the correlation between the given node and its neighbors within the range of $(0, 1)$. This range offers greater flexibility compared to summation (Xu et al., 2019) and enhances generalization capabilities.

**Theorem 3.** The expressive capacity of MPNN-based Graph Transformer using *sigmoid* as attentive function can be as powerful as WL test.

In comparison to *softmax* and *tanh*, *sigmoid* is a more suitable choice for the activation function in attention mechanisms. The key to injectiveness lies in selecting a mapping function $g(\cdot)$ which guarantees $g\left(Q^t(u)K_v^{t\top}(u)\right)V_v^t(u)$ is injective. With this theoretical foundation, we introduce a Graph Transformer with enhanced expressive capacity, namely Expressive Graph Transformer (EGT).

## 4.4 EXPRESSIVE GRAPH TRANSFORMER

Here, we provide a detailed introduction to the proposed EGT. The primary distinction between EGT and vanilla Graph Transformer lies in the activation function used in the attention mechanism and the introduction of learnable parameters in the Feed-Forward Network (FFN). Additionally, we incorporate several techniques to facilitate smoother convergence and enhance overall performance. The structure of EGT is illustrated in Figure 2.

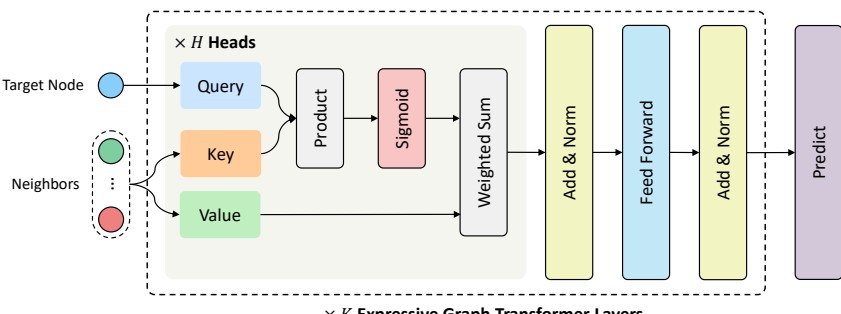

Figure 2: The framework of the Expressive Graph Transformer (EGT). EGT aims to extract information from nodes and neighbors with higher expressive capacity in a message passing manner.

Similar to Graph Transformer, EGT adopts a message passing architecture to progressively accumulate information from the neighborhood. In EGT, each layer $t$ is responsible for capturing information at the $t$-th hop. Given the representation of node $u$ and its neighbors at the $t$-th iteration, we transform them into queries $\boldsymbol{q}^t(u)$, keys $\boldsymbol{k}^t(u)$, and values $\boldsymbol{v}^t(u)$, respectively. When computing attention coefficients $\boldsymbol{w}_v^t(u)$, we employ the *sigmoid* activation function.

$$\boldsymbol{w}_v^t(u) = sigmoid\left(\frac{\boldsymbol{q}^t(u)\boldsymbol{k}_v^{t\top}(u)}{\sqrt{d_h}}\right). \tag{5}$$

The aggregated neighbor representation can be computed by $\boldsymbol{a}^t(u) = \overset{H}{\underset{}{\|}} \left(\sum_{v\in\mathcal{N}(u)} \boldsymbol{w}_v^t(u)\boldsymbol{v}_v^t(u)\right)$. For the combination step, we utilize a feed-forward layer with layer normalization and a non-linear activation function to extract more profound representations. The combination process can be formulated as:

$$\tilde{\boldsymbol{f}}^t(u) = \text{FFN}^t\left(\text{Norm}\left((1+\epsilon_1^t)\boldsymbol{f}^{t-1}(u) + \boldsymbol{a}^t(u)\right)\right),$$
$$\boldsymbol{f}^t(u) = \text{Norm}\left((1+\epsilon_2^t)\boldsymbol{f}^{t-1}(u) + \tilde{\boldsymbol{f}}^t(u)\right). \tag{6}$$

In the combination process, we introduce learnable parameters $\epsilon_1^t$ and $\epsilon_2^t$ to distinguish each node and enhance the high expressive capacity of EGT. Similar to all MPNNs, EGT can be stacked for $K$ layers to integrate information from $K$-hop neighbors. All of these computations can be performed in the form of sparse matrices to enhance efficiency.

**Corollary 4.** EGT has the same discriminative power as WL test.

The node representations at the final layer of EGT are passed through a linear layer to make the final decisions. The model is supervised using a classification loss function to update the parameters.

## 5 EXPERIMENTS

### 5.1 GRAPH CLASSIFICATION

**Datasets.** We use four datasets from GIN (Xu et al., 2019) for graph classification: MUTAG, PTC, PROTEINS, and NCI1 (Debnath et al., 1991; Yanardag & Vishwanathan, 2015). A summary of dataset statistics can be found in Table 1.

Table 1: Statistics of graph classification datasets.

|                | MUTAG | PTC  | PROTEINS | NCI1 |
|----------------|-------|------|----------|------|
| Graphs         | 188   | 344  | 1113     | 4110 |
| Avg. Nodes     | 17.9  | 25.5 | 39.1     | 29.8 |
| Avg. Relations | 57.5  | 77.5 | 184.7    | 94.5 |
| Node Feature   | 7     | 19   | 3        | 37   |
| Classes        | 2     | 2    | 2        | 2    |

**Baseline and evaluation settings.** We evaluate the classification performance by accuracy on these datasets. Consistent with Xu et al. (2019), we employ 5-fold cross-validation to ensure the accuracy and reliability of our results. Our proposed model is compared against four models designed for general graphs (GCN (Kipf & Welling, 2017), GAT (Veličković et al., 2018), GATv2 (Brody et al., 2022), GraphSAGE (Hamilton et al., 2017)), one model with high expressive capacity (GIN (Xu et al., 2019)), and two Graph Transformers (GT (Dwivedi & Bresson, 2021), Graphormer (Ying et al., 2021)). We apply virtual nodes for MPNNs to aggregate information from graphs, following the approach in Hu et al. (2020a). General settings, including the embedding layer, hidden dimensions, etc., remain consistent across all models to ensure a fair comparison. The number of hops for all models is set to 4.

Table 2: Performance comparison of accuracy on graph classification

|                    | MUTAG      | PTC        | PROTEINS   | NCI1       |
|--------------------|------------|------------|------------|------------|
| GCN                | 96.22%     | 86.77%     | 79.64%     | 88.73%     |
| GAT                | 96.22%     | 86.77%     | 79.28%     | 91.89%     |
| GATv2              | 97.30%     | 87.02%     | 80.18%     | 92.20%     |
| GraphSAGE          | 95.68%     | 90.88%     | 79.10%     | 89.39%     |
| GIN                | 97.30%     | 93.82%     | 83.33%     | 92.29%     |
| GT                 | 96.22%     | 92.94%     | 80.18%     | 88.52%     |
| Graphormer         | 95.68%     | 80.29%     | 83.15%     | 83.70%     |
| $\text{EGT}_{softmax}$ | 96.22%     | 94.41%     | 80.81%     | 91.90%     |
| $\text{EGT}_{tanh}$    | 97.30%     | 94.71%     | 85.68%     | 92.92%     |
| EGT                | **98.38%** | **96.18%** | **86.76%** | **96.47%** |

**Result.** Table 2 provides an overview of performance comparisons on graph classification datasets. With the virtual node framework, the graph classification capability of MPNNs is significantly enhanced compared to Xu et al. (2019). It even surpasses Graphormer on the PTC and NCI1 datasets, despite Graphormer can be more expressive than the WL test. Thanks to dynamic attention, GATv2 outperforms GAT, but still falls short when compared to GIN due to limited expressive power. As a GNN with high expressive capacity, GIN surpasses all other baselines. However, EGT offers better convergence with the help of FFN and normalization. EGT consistently exhibits the best performance among the baselines in all cases, confirming its effectiveness in graph classification.

**Ablation study.** To further investigate the improvements stemming from the activation function, we replace the $sigmoid$ in EGT with $softmax$ and $tanh$, denoted as $\text{EGT}_{softmax}$ and $\text{EGT}_{tanh}$, respectively. With $softmax$, there is a significant decrease in accuracy and the model degenerate to the architecture similar to GT. The results from $tanh$ is better than $softmax$ since it can recognize the neighbor counts. However, its expressivity is limited as well.

## 5.2 NODE CLASSIFICATION

**Datasets.** The node classification experiments encompass a range of popular datasets, including the citation network datasets (Pubmed, Cora) (Yang et al., 2016), coauthor datasets (CS, Physics) (Shchur et al., 2018), co-purchase datasets (Computer, Photo) (McAuley et al., 2015), and the image category dataset (Flickr) (Zeng et al., 2019). Statistical details of these datasets are presented in Table 3.

Table 3: Statistics of node classification datasets.

|              | Pubmed | Cora   | CS      | Physics | Computer | Photo   | Flickr  |
|--------------|--------|--------|---------|---------|----------|---------|---------|
| Nodes        | 19,717 | 2,708  | 18,333  | 34,493  | 13,752   | 7,650   | 89,250  |
| Relations    | 88,651 | 10,556 | 163,788 | 495,924 | 491,722  | 238,163 | 899,756 |
| Node Feature | 500    | 1,433  | 6,805   | 8,415   | 767      | 745     | 500     |
| Classes      | 3      | 7      | 15      | 5       | 10       | 8       | 7       |

**Baseline and evaluation settings.** We partition the datasets into training, validation, and test sets, following a ratio of 60%, 20%, 20%, respectively. Each experiment is repeated three times with different random seeds to ensure the credibility of the results. In contrast to graph classification, we replace Graphormer with the latest state-of-the-art work on node classification, NAGphormer (Chen et al., 2023), for comparison. The number of neighbor hops is set to 2.

**Result.** Table 4 displays the performance of baseline methods and our proposed approach in the task of node classification, establishing a new state-of-the-art. Among the baseline models, NAGphormer, a Graph Transformer that treats each node and its neighbors as a sequence, outperforms other baseline models and highlights the effectiveness of Graph Transformers. Among all MPNNs, GT demonstrates the most competitive performance. However, the architectural and attention function enhancements in EGT lead to improved results.

**Ablation study.** When comparing EGT with $EGT_{softmax}$ and $EGT_{tanh}$, it becomes evident that *sigmoid* outperforms *softmax* and *tanh* in most cases, underscoring the significance of fully capturing information from neighbors in node classification. In node classification scenarios, information from node features holds greater importance. Therefore, the performance gap between EGT and its variants is relatively smaller compared to the gap observed in graph classification. Besides, due to $EGT_{tanh}$ cannot differentiate sign of features, it may display poor performance in some scenarios, like the performance on the Cora dataset.

Table 4: Performance comparison of accuracy on node classification

|                 | Pubmed     | Cora       | CS         | Physics    | Computer   | Photo      | Flickr     |
|-----------------|------------|------------|------------|------------|------------|------------|------------|
| GCN             | 83.29%     | 80.38%     | 91.86%     | 95.66%     | 87.40%     | 93.16%     | 50.53%     |
| GAT             | 83.48%     | 81.67%     | 91.30%     | 95.34%     | 88.48%     | 93.59%     | 49.36%     |
| GATv2           | 83.62%     | 81.24%     | 91.46%     | 95.70%     | 89.97%     | 94.05%     | 50.38%     |
| GraphSAGE       | 85.45%     | 84.47%     | 91.63%     | 95.64%     | 86.22%     | 92.53%     | 50.36%     |
| GIN             | 87.33%     | 84.19%     | 91.48%     | 95.84%     | 86.32%     | 93.40%     | 50.03%     |
| GT              | 86.90%     | 84.13%     | 94.00%     | 95.92%     | 89.45%     | 94.75%     | 51.86%     |
| NAGphormer      | 88.08%     | 84.26%     | 94.69%     | 96.65%     | 89.05%     | 94.81%     | 52.13%     |
| $EGT_{softmax}$ | 88.16%     | **85.08%** | 95.56%     | 97.03%     | 90.03%     | 94.88%     | 51.41%     |
| $EGT_{tanh}$    | **88.78%** | 77.49%     | 95.62%     | **97.04%** | 90.31%     | 94.66%     | 52.03%     |
| EGT             | 88.63%     | 84.93%     | **96.73%** | 96.92%     | **90.54%** | **95.64%** | **52.40%** |

**Result when downsample neighbors.** What happens when we downsample neighbors instead of performing full sampling in EGT? In this scenario, the information related to the number of neighbors is weakened because we only select a predetermined number of neighbors. We assess the performance of EGT with up to 5 one-hop neighbors and 15 two-hop neighbors. The remarkable results are presented in Table 5.

From Table 5, we can observe a performance decline across all models when downsampling neighbors. Surprisingly, there does not exist pronounced change of EGT in performance than the others.

Table 5: Performance comparison of accuracy on node classification in the downsampling setting

|  | Pubmed | Cora | CS | Physics | Computer | Photo | Flickr |
|---|---|---|---|---|---|---|---|
| GCN | 82.10% | 79.46% | 90.99% | 95.50% | 86.97% | 92.75% | 48.28% |
| GAT | 82.22% | 81.49% | 91.32% | 95.12% | 86.31% | 92.37% | 48.51% |
| GATv2 | 82.49% | 81.18% | 91.30% | 95.63% | 87.16% | 92.66% | 48.83% |
| GraphSAGE | 84.91% | 83.73% | 91.87% | 95.52% | 85.99% | 91.96% | 48.84% |
| GIN | 86.98% | 83.61% | 91.92% | 95.88% | 86.70% | 92.83% | 49.30% |
| GT | 83.93% | 83.76% | 93.91% | 96.77% | 89.17% | 94.14% | 50.32% |
| NAGphormer | 87.08% | 82.66% | 94.34% | 96.64% | 87.77% | 93.92% | 49.38% |
| $\text{EGT}_{softmax}$ | 88.14% | **85.06%** | 95.44% | 97.05% | 89.82% | 94.90% | 50.24% |
| $\text{EGT}_{tanh}$ | **88.61%** | 77.12% | 95.58% | 97.01% | 89.81% | 94.77% | 50.82% |
| EGT | 88.49% | 84.89% | **95.90%** | **97.08%** | **90.12%** | **95.42%** | **50.92%** |

This phenomenon can be attributed to the $sigmoid$ function's ability to allocate arbitrary weighted sums to neighbors, whereas $softmax$ restricts the sum to be 1. Consequently, $sigmoid$ is more flexible than $softmax$ even if the neighbors counts of nodes are the same. Additionally, $softmax$ may exhibits unstable behavior when it comes to differently sampled neighbors, whereas $sigmoid$ is not affected.

## 5.3 ROBUSTNESS TO NOISE

We examine the robustness of EGT to noise to validate the superiority from considering number of neighbors. Specifically, we focus on structural noise following the approach in GATv2 (Brody et al., 2022): given a graph $\mathcal{G} = (\mathbb{V}, \mathbb{E})$ and a noise ratio $0 \leq p < 1$, we randomly sample $|\mathbb{E}| \times p$ non-existing edges $\mathbb{E}'$ from $\mathbb{V} \times \mathbb{V} \setminus \mathbb{E}$. Subsequently, we train the GNN on the noisy graph $\mathcal{G}' = (\mathbb{V}, \mathbb{E} \cup \mathbb{E}')$. We experiment with various noise levels $p \in 0, 0.1, 0.2, 0.3, 0.4$ on the Cora, Computer, and Flickr datasets. We compare the results with competitive baseline models (GATv2, GraphSAGE, GT, and NAGphormer).

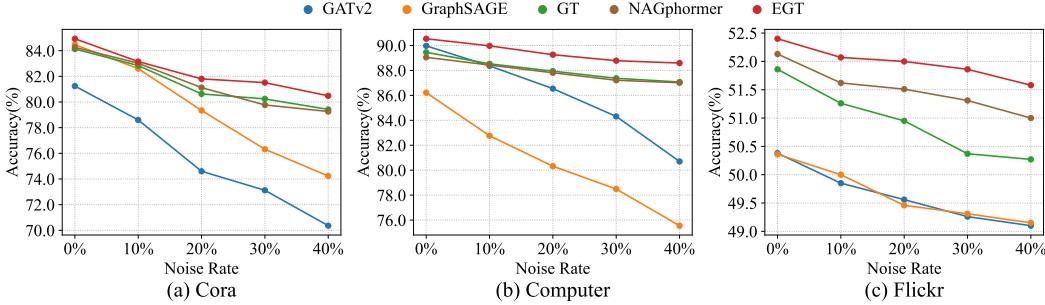

Figure 3: The performance when it comes to structural noise on node classification

The results in Figure 3 reveal that all models experience natural declines in accuracy as $p$ increases. However, EGT demonstrates a milder degradation compared to the baseline models, underscoring the effectiveness of modifying the attention function. Among the baselines, GT and NAGphormer exhibit substantial robustness, thanks to the Transformer architecture. When compared to GT and NAGphormer, EGT exhibits more significant improvements as $p$ increases, particularly on the challenging Flickr dataset.

In summary, we conclude that $softmax$ is more susceptible to noise due to its normalized nature. The attention results of $softmax$ are more likely to be dominated by extreme noise edges. Conversely, since $sigmoid$ assigns individual weights to each edge, it proves to be more robust to noise. Although GATv2 exhibits better stability than GAT in noisy settings, as verified in Brody et al. (2022), it does not outperform other stronger baseline models and maintains similar performance to GraphSAGE.

## 5.4 Hyperparameter Study

We further evaluate the performance of EGT on Cora, Computer and Flickr dataset with respect to the number of Transformer layers, hidden dimension and number of heads in the attention mechanism to evaluate the sensitivity.

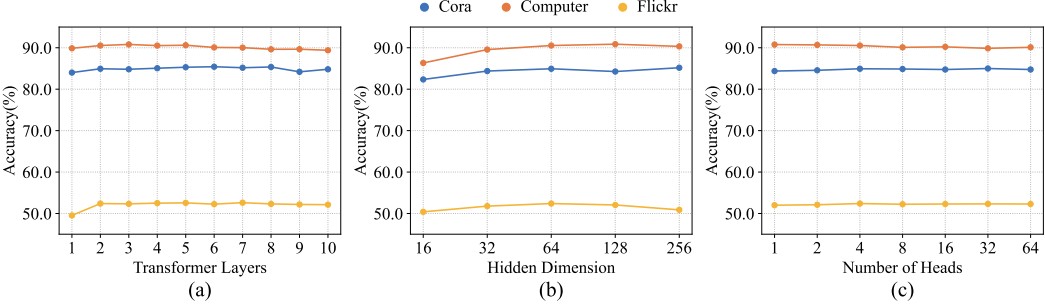

Figure 4: Sensitivity analysis for different Transformer layers, hidden dimensions, and the number of attention heads on node classification

We conducted experiments with varying numbers of layers, ranging from 1 to 10, while simultaneously adjusting the number of neighbor hops. The results in Figure 4(a) reveal that EGT exhibits stability concerning the number of Transformer layers. However, a notable drop in performance is observed when using only 1 layer on the Flickr dataset, indicating that insufficient information is captured in this scenario. Next, we investigated the impact of the hidden dimension by evaluating various hidden dimension values, ranging from 16 to 256. The results in Figure 4(b) demonstrate that performance initially improves with an increase in hidden dimension. However, if the hidden dimension becomes excessively large, it may lead to convergence challenges. Regarding the number of heads in the attention mechanism, we varied the head counts from 1 to 64, and the corresponding results are presented in Figure 4(c). EGT exhibits robustness to changes in the number of heads.

It's worth noting that the results of these experiments surpass those listed in Table 4, indicating that the performance of EGT can be further enhanced through parameter fine-tuning. Overall, these experiments shed light on the factors that influence the performance of EGT, providing insights into optimizing configurations.

## 5.5 Discussion about the expressive power of GATs

Apart from GT, GAT and GATv2 are also attentive MPNNs. As discussed in Brody et al. (2022), GAT employs static attention, where the ranking of attention coefficients is solely determined by the neighbors and is not conditioned on the query node. On the other hand, GATv2 utilizes dynamic attention, making it strictly more expressive than GAT. Nevertheless, due to the restriction imposed by $softmax$, both GAT and GATv2 cannot judge the number of neighbors accurately, resulting in lower expressiveness than the WL test.

## 6 Conclusion

In this paper, we identify a limitation in the expressive capacity of the attention mechanism used in Graph Transformers. To overcome this constraint, we introduce the Expressive Graph Transformer (EGT), which exhibits the same level of expressive power as the WL test. Our comprehensive evaluations on graph classification and node classification consistently demonstrate the effectiveness of EGT. Through additional analyses, we show that EGT displays robustness to structural noise and hyperparameters. EGT is poised to become a new benchmark for the graph classification and node classification task, and its simplicity in architecture allows for easy extension to other tasks such as link prediction. We encourage the community to adopt EGT as an alternative to Graph Transformers and consider using the *sigmoid* activation function instead of *softmax* in attentive MPNNs.

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

## A    PROOF FOR THEOREM 1

*Proof.* Prior to delving into the expressiveness of the attentive function, we first discuss the prerequisites for an MPNN to attain the upper bound of the expressiveness equivalent to the WL test, as outlined by Xu et al. (Xu et al., 2019):

- The MPNN aggregates and updates all nodes and their neighbors iteratively by AGGREGATE and COMBINE operations which must be injective.
- The graph-level readout on the multiset of node features must be injective.

Therefore, to achieve expressive power as WL test, it is crucial for the AGGREGATE and COMBINE operations in Graph Transformers to be injective.

Now, let's proceed to prove the Theorem 1. Given a non-injective function as an attentive function, there exists different $v$ ensuring equal $\boldsymbol{w}_v^t(u)$. prove that the *softmax* activation function is not injective by presenting a counterexample. Consider two nodes, $u$ and $u'$, along with their respective sets of neighbors, $\mathcal{N}(u)$ and $\mathcal{N}(u')$. We assume that $2 \cdot \mathrm{card}(\mathcal{N}(u)) = \mathrm{card}(\mathcal{N}(u'))$, and for any node $v \in \mathcal{N}(u)$, there exist two nodes, $v_1', v_2' \in \mathcal{N}(u')$ such that $\boldsymbol{f}^0(v) = \boldsymbol{f}^0(v_1') = \boldsymbol{f}^0(v_2')$, where $\boldsymbol{f}^0(\cdot)$ represents the initial node features, and $\mathrm{card}(\cdot)$ denotes the number of elements in the multiset.

Under these conditions, it follows that for every node $v$, there exists two nodes $v_1'$ and $v_2'$ such that $\boldsymbol{w}_v^1(u) = 2 \cdot \boldsymbol{w}_{v_1'}^1(u') = 2 \cdot \boldsymbol{w}_{v_2'}^1(u')$ where $\boldsymbol{w}_v^1(u), \boldsymbol{w}_{v_1'}^1(u'), \boldsymbol{w}_{v_2'}^1(u')$ are *softmax* attention weights as defined in Equation 4. Furthermore, the aggregated results of $\mathcal{N}(u)$ and $\mathcal{N}(u')$ are identical. In summary, the AGGREGATE operation is not injective when the attentive function is *softmax*, thereby establishing the purposed theorem.

## B    PROOF FOR THEOREM 2

*Proof.* Before proving the theorem, we first illustrate that the attention mechanism with *tanh* fails to differentiate between positive and negative inputs. In this scenario, the computation of each neighbor in the AGGREGATE function follows:

$$
\begin{aligned}
\boldsymbol{w}_v^t(u)\boldsymbol{v}_v^t(u) &= \tanh\Big(\boldsymbol{q}^t(u)\boldsymbol{k}_v^{t\top}(u)\Big)\boldsymbol{v}_v^t(u) \\
&= \tanh\Big(\boldsymbol{q}^t(u)\big(\boldsymbol{W}_K^t\boldsymbol{f}^{t-1}(v)\big)^\top\Big)\boldsymbol{W}_V^t\boldsymbol{f}^{t-1}(v).
\end{aligned}
\tag{7}
$$

Here we omit $1/\sqrt{d_h}$ for brevity. Since $\boldsymbol{W}_K^t\boldsymbol{f}^{t-1}(v)$ and $\boldsymbol{W}_V^t\boldsymbol{f}^{t-1}(v)$ are linear projections, we can conclude that $\boldsymbol{W}_K^t\boldsymbol{f}^{t-1}(v') = -\boldsymbol{W}_K^t\boldsymbol{f}^{t-1}(v)$ and $\boldsymbol{W}_V^t\boldsymbol{f}^{t-1}(v') = -\boldsymbol{W}_V^t\boldsymbol{f}^{t-1}(v)$ for any $v$ where $\boldsymbol{f}^{t-1}(v') = -\boldsymbol{f}^{t-1}(v)$. According to the nature of *tanh* that $\tanh(x) = -\tanh(-x)$, we have

$$
\begin{aligned}
\boldsymbol{w}_v^t(u)\boldsymbol{v}_v^t(u') &= \tanh\Big(\boldsymbol{q}^t(u)\big(\boldsymbol{W}_K^t\boldsymbol{f}^{t-1}(v')\big)^\top\Big)\boldsymbol{W}_V^t\boldsymbol{f}^{t-1}(v') \\
&= -\tanh\Big(-\boldsymbol{q}^t(u)\big(\boldsymbol{W}_K^t\boldsymbol{f}^{t-1}(v)\big)^\top\Big)\boldsymbol{W}_V^t\boldsymbol{f}^{t-1}(v) \\
&= \tanh\Big(\boldsymbol{q}^t(u)\big(\boldsymbol{W}_K^t\boldsymbol{f}^{t-1}(v)\big)^\top\Big)\boldsymbol{W}_V^t\boldsymbol{f}^{t-1}(v).
\end{aligned}
\tag{8}
$$

Thus, attention mechanism with *tanh* remains agnostic to positive and negative inputs. Consequently, The attention is not injective and lead to a non-injective AGGREGATE.

## C    PROOF FOR THEOREM 3

*Proof.* We begin by presenting evidence for the injectiveness of the attention mechanism with *sigmoid*. The function $sigmoid(x) = \dfrac{1}{1 + \exp(-x)}$ maps any value $x$ to a positive range, implying

that for any node $u$ and its neighbor $v$, $w_v^t(u)$ is positive. Suppose that there exists a neighbor $v' \neq v$ such that $\boldsymbol{w}_v^t(u)\boldsymbol{v}_v^t(u) = \boldsymbol{w}_{v'}^t(u)\boldsymbol{v}_{v'}^t(u)$ for any $\boldsymbol{W}_Q^t, \boldsymbol{W}_K^t, \boldsymbol{W}_V^t$, we can obtain $\boldsymbol{v}_v^t(u) = \boldsymbol{v}_{v'}^t(u) = 0$ or $\dfrac{\boldsymbol{w}_v^t(u)}{\boldsymbol{w}_{v'}^t(u)} = \dfrac{\boldsymbol{v}_{v'}^t(u)}{\boldsymbol{v}_v^t(u)}$. Since $\boldsymbol{v}_v^t(u)$ is linear projection, we can conclude that the $v$ and $v'$ have the same sign. Considering both the linear projection and *sigmoid* function are monotonic, for $\boldsymbol{q}^t(u) > 0$, $\boldsymbol{W}_K^t = \boldsymbol{W}_V^t \neq 0$ and $v > v'$, there must exist $\dfrac{\boldsymbol{w}_v^t(u)}{\boldsymbol{w}_{v'}^t(u)} > 1$ and $\dfrac{\boldsymbol{v}_{v'}^t(u)}{\boldsymbol{v}_v^t(u)} < 1$. Hence we have reached a contradiction. Thus, there exists $\boldsymbol{W}_Q^t, \boldsymbol{W}_K^t, \boldsymbol{W}_V^t$ ensuring that attention mechanism with *sigmoid* is injective.

The set of neighbors can be regarded as a multiset with a bounded size. Since the size of node features is countable, there exists a function that maps neighbor nodes to natural numbers. Moreover, given the bounded cardinality of the neighbor multiset, there exists a function that ensures $\sum_{v \in \mathcal{N}(u)} w_v^t(u) V_v^t(u)$ is unique, thereby proving the injectivity of AGGREGATE.

For COMBINE, we modify the vanilla function in the graph transformer by introducing a parameter $\gamma$. The modified function is defined as follows:

$$\text{COMBINE}^t = \text{FFN}^t\big((1+\gamma)f^{t-1}(u) + a^t(u)\big). \tag{9}$$

Following Xu et al. (Xu et al., 2019), when $\gamma$ is irrational, there exists a function such that $(1+\gamma)f^{t-1}(u) + \sum_{v \in \mathcal{N}(u)} f^{t-1}(v)$ is unique. Since FFN can work as a universal approximator (Hornik et al., 1989; Yun et al., 2020) used to model and learn the composition of functions, our modified combination function can be injective as well.

Hence, by using multi-layer perceptrons to learn a graph-level readout, attention with the *sigmoid* activation function can achieve the expressiveness of the WL test.

## D   PROOF OF COROLLARY 4

*Proof.* We know that the expressive capacity of a MPNN can be as powerful as WL test when its AGGREGATE and COMBINE operations are injective. As proved in Theorem 3, the COMBINE with $sigmoid$ in EGT is injective. Now, let's consider the entire combination process in EGT, which includes $(1 + \epsilon_1^t)\boldsymbol{f}^{t-1}(u) + \boldsymbol{a}^t(u)$. This combination process can also be regarded as injective because it involves both the irrational $\epsilon_1^t$ and the aggregated neighbor representation $\boldsymbol{a}^t(u)$, which are both unique and distinct for each node and its neighbors. Consequently, we can conclude that the expressive power of EGT is equal to that of the WL test.

## E   IMPLEMENTATION DETAILS

All experiments of this model are carried out on a Linux server equipped with one RTX 3090 GPU. PyTorch 1.12.0 and DGL 1.1.1 deep learning libraries is applied to build and train our neural network. The hidden size $d$ of all modules is 64. Adam optimizer (Kingma & Ba, 2014) is applied while training the model. On graph classification, we train models for 100 epochs by batch size of 256 and learning rate of 0.005 with scheduler changing learning rate to 0.0025 at epoch 50. On node classification, we train models for 100 epochs by batch size of 65,536 and learning rate of 0.001. To evaluate precisely, we conduct five-fold cross-validation on graph classification and repeat every experiment 3 times with different random seeds on node classification. For implementing Graphormer and NAGphormer, we use 2 layers of encoder for them, which is the best result by trying multiple values. The NAGphormer is not suitable for graph classification. The number of heads of all attentive models (GAT, GATv2, GT, Graphormer, NAGphormer, and EGT) are all 4 to compare fairly.

## F   COMPUTATION COMPLEXITY AND SCALABILITY

When it comes to computation complexity and scalability, EGT has the similar complexity to MPNNs, which is linear, as it involves aggregating information from neighboring nodes iteratively.

For scalability, EGT does not compute statistics like Graphormer, which requires computing global statistics across the entire graph. This means that EGT can be more scalable across different graphs, as it doesn't depend on calculating global statistics that might become prohibitively expensive for large graphs.

Overall, EGT's computational complexity and scalability characteristics make it a suitable choice for a wide range of graph-based tasks, particularly when dealing with large and diverse graphs.

