# OpenReview forum: "Activation Function Matters in Graph Transformers"
_ICLR.cc/2024/Conference — ICLR 2024 Conference Withdrawn Submission_

### Official Review · Reviewer_w1tJ · 2023-10-27

**Soundness:** 1 poor
**Presentation:** 2 fair
**Contribution:** 1 poor
**Rating:** 3
**Confidence:** 4

**Summary:**

This work studies the effect of the activation function for attention mechanisms in the message-passing neural network(MPNN)-based Graph Transformers.
Comparing the original softmax with the sigmoid and tanh functions, the authors demonstrate that the sigmoid function reaches the most powerful expressiveness in graph isomorphism tasks, as powerful as the WL algorithm.
Theoretically, they show that sparse attention with sigmoid is a sum-aggregation which is more expressive than sparse attention with softmax, which is actually a mean-aggregation. They also illustrate that the negative value in Tanh will result in suboptimal expressive power.
The authors verify their findings empirically on several graph and node-level benchmarks, in comparison with several MPNNs and graph transformers.
On graph-level tasks, the variant with the sigmoid function reaches the best performance in general.
On node-level tasks, the outperformance is less remarkable.

**Strengths:**

1. This work is well-written in general.
2. The authors conducted several good analysis experiments such as the robustness to noise and hyperparameter study.

**Weaknesses:**

1. This work focuses on 'MPNN-based Graph Transformers' which is a controversial concept. Even though previous works in graph transformers acknowledge the contribution of the pioneer works with sparse attention, such as Dwivedi & Bresson (2021),
most researchers concede that the key difference between Graph Transformers and attention-based MPNNs is the ability to aggregate beyond first-hop neighborhoods, in order to address over-squashing issues and expressive-limitation (bounded by 1-WL algorithms).
(Ying et al., 2021; Kreuzer et al., 2021; Rampasek et al., 2022; Zhang et al., 2023; Ma et al., 2023)
2. Even though many recent works on graph transformers focus on computational efficiency (mainly for node-level tasks), there are also many recent graph transformers emphasizing expressive power, neglected by the authors (Kreuzer et al., 2021; Chen et al., 2022; Rampasek et al., 2022; Zhang et al., 2023; Ma et al., 2023).
3. Previous works have introduced degree-scalers to mitigate the information loss of degrees in mean-aggregation (including softmax-based attention), i.e., PNA (Corso et al., 2020) for MPNNs, GRIT (Ma et al., 2023) for graph transformers. These techniques are not mentioned or compared in this work.
4. The work does not conduct experiments on several more widely used graph-level benchmarks in graph transformer literature, e.g., BenchmarkingGNNs (Dwivedi et al., 2020) and Long-Range Graph Benchmarks (Dwivedi et al., 2022). Datasets, like MUTAG, PTC, etc, are rarely used for evaluating graph transformers in recent works.
5. The authors do not include any latest graph transformers after 2021 in the experiments of graph-level tasks, including but not limited to Rampasek et al., (2022), Zhang et al., (2023), and Ma et al., (2023), which have renewed SOTA performances on several graph-level tasks. (They do include NAGPhormer (2023) for node-level tasks)

---------------

- Dwivedi, V. P. and Bresson, X. A Generalization of Transformer Networks to Graphs. In Proc. AAAI Workshop Deep Learn. Graphs: Methods Appl., 2021.
- Ying, C., Cai, T., Luo, S., Zheng, S., Ke, G., He, D., Shen, Y., and Liu, T.-Y. Do Transformers Really Perform Badly for Graph Representation? In Adv. Neural Inf. Process. Syst., 2021.
- Kreuzer, D., Beaini, D., Hamilton, W. L., L ́ etourneau, V., and Tossou, P. Rethinking Graph Transformers with Spectral Attention. In Adv. Neural Inf. Process. Syst., 2021
-Rampasek, L., Galkin, M., Dwivedi, V. P., Luu, A. T., Wolf, G., and Beaini, D. Recipe for a General, Powerful, Scalable Graph Transformer. In Adv. Neural Inf. Process. Syst., 2022.
- Zhang, B., Luo, S., Wang, L., and Di, H. Rethinking the expressive power of gnns via graph biconnectivity. In Proc. Int. Conf. Learn. Representations, 2023.
- Ma, L., Lin, C., Lim, D., Romero-Soriano, A., Dokania, P.K., Coates, M., Torr, P. &amp; Lim, S. Graph Inductive Biases in Transformers without Message Passing. In Proc. Int. Conf. Mach. Learn., 2023.
- Bresson, X. and Laurent, T. Residual Gated Graph ConvNets. arXiv, 2018
- Corso, G., Cavalleri, L., Beaini, D., Lio, P., and Velickovic, P. Principal Neighbourhood Aggregation for Graph Nets. In Adv. Neural Inf. Process. Syst., 2020
- Dwivedi, V. P., Joshi, C. K., Laurent, T., Bengio, Y., and Bresson, X. Benchmarking Graph Neural Networks. arXiv:2003.00982 [cs, stat], July 2020.
- Dwivedi, V. P., Rampasek, L., Galkin, M., Parviz, A., Wolf, G., Luu, A. T., and Beaini, D. Long Range Graph Benchmark. In Adv. Neural Inf. Process. Syst. (Dataset Track), 2022.
- Chen, D., O’Bray, L., and Borgwardt, K. Structure-Aware Transformer for Graph Representation Learning. In Proc. Int. Conf. Mach. Learn., pp. 3469–3489, 2022.

**Questions:**

As mentioned in the weaknesses.

---

### Official Review · Reviewer_1n8G · 2023-10-31

**Soundness:** 2 fair
**Presentation:** 3 good
**Contribution:** 3 good
**Rating:** 5
**Confidence:** 3

**Summary:**

The authors study an important problem, the activation function of Graph Transformers (GTs) and proposed an enhanced variant of GT (EGT). EGT utilizes sigmoid instead of softmax as activation functions. The effectiveness of EGT is theoretically and empirically proved.

**Strengths:**

S1. The studied problem, i.e. activation function of GT, is a vital yet less explored problem.

S2. Theoretical contributions of expressiveness of GTs.

S3. Good performance proved by extensive experiments.

**Weaknesses:**

W1. GTs are fundamentally different from GNNs where the core assumption is that all nodes are aggregated, i.e. $v \in \mathbb{V}$ instead of $v \in \mathcal{N}(u)$ in Eq.3. With that in mind, the claim that softmax cannot effectively discern the number of neighbors is questionable (because the target node is also considered). The example in Figure 1 (a) is also incorrect, considering a node u, with identical neighbours, say $x_u=2, x_v=1, \forall v\in \mathcal{N}_1(u)$. For different neighborhood set  $\mathcal{N}_2(u)=(v_1, v_2)$  (Figure 1 (a) left) $\mathcal{N}'(u)=(v_1, v_2, v_3, v_4)$  (right), $softmax(x_u \cup \mathcal{N}(u) )$ leads to $(0.58, 0.21, 0.21)$, and $(0.4, 0.15, 0.15, 0.15, 0.15)$. Note the distributions are different: the target node u (the first one) gets less attention when the number of neighbors increases.

W2. The authors claim that: "softmax retains its effectiveness when the number of computed components remains constant, as seen in applications like image or natural language processing" This is not true according to my understanding, for NLP, repeating words (tokens) shifts the attention distribution and the overall distribution. Empirically, for a trained LM,  repeating words largely shifts the distribution, leading to out-of-distribution problems for LMs, which also demonstrates the effectiveness of softmax in discerning the number of neighbors.

W3. The notations confusing: (1) $\boldsymbol{v}_v^t(u)$ $\boldsymbol{k}_v^t(u)$ are functions of u, yet has nothing to do with node u; (2) typo of $\boldsymbol{W}_V^t$ v.s $W_v^t$ behind Eq. 3.

W4. Citation of GIN is needed for the design of $\epsilon$ in Eq. 6.

W5. There is already an EGT [1].


[1] Hussain et al. Global Self-Attention as a Replacement for Graph Convolution. In KDD2022.

**Questions:**

Q1: Clarify W1 please.

Q2: Clarify W4.

---

### Official Review · Reviewer_TK6L · 2023-11-01

**Soundness:** 2 fair
**Presentation:** 3 good
**Contribution:** 2 fair
**Rating:** 5
**Confidence:** 3

**Summary:**

This paper investigates the impact of different activation functions in graph neural networks (GNNs). The authors conducted a theoretical analysis on the expressive capabilities of three activation functions, including softmax, tanh, and sigmoid. Based on the conclusions drawn from the analysis, they improved previous methods and conducted experiments on various datasets to validate their proposal. The experimental results demonstrate the effectiveness of their strategy.

**Strengths:**

a. The motivation is interesting. The authors discovered that different activation functions have varied impacts on the information propagation process in GNNs.
b. The experimental results are impressive. The authors managed to outperform the original baseline models across multiple datasets by simply changing the activation function, which seems to be a straightforward and effective strategy.

**Weaknesses:**

a. Some crucial proposals lack clear explanations. For example, the two learnable parameters in equation (6) – how do they achieve the functionality of distinguishing each node as stated in the paper? In addition, the ablation study is missing an analysis of the newly introduced parameters. Without these two learnable parameters, and by simply changing the activation function, what would be the effect on the model's performance?

b. The discussion on activation functions is insufficient. Other commonly used activation functions, such as ReLU, seem capable of addressing the problem about insensitivity to the neighbor number raised in the paper. Have the authors conducted any theoretical analysis or experimental attempts with these functions?

**Questions:**

a. Regarding the experimental results, there are some doubts. Following the paper's conclusion, sigmoid as an activation function should be superior to softmax and tanh. Why is it that in Table 4, the results on some datasets are worse with sigmoid?

---

### Official Review · Reviewer_Ryu8 · 2023-11-07

**Soundness:** 3 good
**Presentation:** 3 good
**Contribution:** 2 fair
**Rating:** 5
**Confidence:** 4

**Summary:**

Activation function lies at the key part of transformer layer. In most cases, Softmax function is applied to the attention weight matrix in order to normalize the neighbor distribution. However, it's not always the perfect choice as indicated by recent work. In this work, authors study the potential impact on the expressivity of graph transformer in distinguishing different graph structures. From the given theorem, we can see that both Softmax and Sigmoid function are still injective after being applied the attention weight, but the situation is different when using hyperbolic tangent function. The presentation of this work is overall good to make it easy to get the main idea. The modification of activation function seems to be practical on improving the performance of graph transformer on classification performance.

**Strengths:**

Strengths:
1. This work studies the impact of activation functions, mainly about softmax, sigmoid, hyperbolic tangent functions.
2. Theoretical analysis gives more insight on the impact of different activation functions on the expressivity of graph transformer.
3. Experiments are conducted on classical benchmarks to demonstrate the effectiveness of modified transformer layer.

**Weaknesses:**

Weaknesses:
1. The discussion on the impact of activation functions are not completed yet. The main consideration of using Softmax function is close to its normalization over the involved nodes. Actually there are many types of activation functions that can be applied. Then why should we only discuss the three exponential functions? As we know that, graph isomorphic network (GIN) derives that just addition operator over the neighbors can be a strong method to improve the expressivity. The properties of linear rectifiers partially keeps the original values. What kind of things can happen if we select them as the activation functions?

2. There is still concern on the capability of using sigmoid as the activation function. We can see that applying sigmoid function to the attention weight equals to approximately creating a binary mask where negative similarity between a pair of nodes will be masked out. In this case, it only focus on node pairs with positive similarity, but ignore the rest of dissimilar neighbors. Let's take a pair of nodes as example. The first node includes 2 positive and 4 negative neighbors, while the second node includes 2 positive and 2 negative neighbors. In such situation, we will get the same value for these two nodes, even they have totally different graph structure.

3. The contributions of each component is not clear enough to demonstrate superiority of modification on the activation function. (i) The transformer layer contains layer normalization, while only applying sigmoid function to attention layer might have gradient exploration issue if the number of neighbors are over a certain number. (2) In this work, it should apply the modification to different base models, but not only to the designed transformer layer in Equation (5) and (6). As we can see that, only changing the activation function is a very general technology. If someone plans to demonstrate its effectiveness, it should keep the rest of part the as the baseline, but only changing the activation function of each layer. Please conduct experiments to make the modification more convincing.

**Questions:**

please refer to mentioned points above.